# Large-Scale Laboratory Experiments on Mussel Dropper Lines in Ocean Surface Waves

Rebekka Gieschen [1,*], Christian Schwartpaul [1], Jannis Landmann [2], Lukas Fröhling [2], Arndt Hildebrandt [2] and Nils Goseberg [1,3]

1 Leichtweiß-Institute for Hydraulic Engineering and Water Resources, Technische Universität Braunschweig, Beethovenstraße 51a, 38106 Braunschweig, Germany; c.schwartpaul@tu-braunschweig.de (C.S.); n.goseberg@tu-braunschweig.de (N.G.)

2 Ludwig-Franzius-Institut for Hydraulic, Estuarine and Coastal Engineering, Leibniz Universität Hannover, Nienburger Str. 4, 30167 Hannover, Germany; landmann@lufi.uni-hannover.de (J.L.); froehling@lufi.uni-hannover.de (L.F.); hildebrandt@lufi.uni-hannover.de (A.H.)

3 Coastal Research Center, Leibniz Universität Hannover and Technische Universität Braunschweig, Merkurstraße 11, 30419 Hannover, Germany

* Correspondence: r.gieschen@tu-braunschweig.de

**Abstract:** The rapid growth of marine aquaculture around the world accentuates issues of sustainability and environmental impacts of large-scale farming systems. One potential mitigation strategy is to relocate to more energetic offshore locations. However, research regarding the forces which waves and currents impose on aquaculture structures in such conditions is still scarce. The present study aimed at extending the knowledge related to live blue mussels (*Mytilus edulis*), cultivated on dropper lines, by unique, large-scale laboratory experiments in the Large Wave Flume of the Coastal Research Center in Hannover, Germany. Nine-months-old live dropper lines and a surrogate of 2.0 m length each are exposed to regular waves with wave heights between 0.2 and 1.0 m and periods between 1.5 and 8.0 s. Force time histories are recorded to investigate the inertia and drag characteristics of live mussel and surrogate dropper lines. The surrogate dropper line was developed from 3D scans of blue mussel dropper lines, using the surface descriptor Abbott–Firestone Curve as quality parameter. Pull-off tests of individual mussels are conducted that reveal maximum attachment strength ranges of 0.48 to 10.55 N for mussels that had medium 3.04 cm length, 1.60 cm height and 1.25 cm width. Mean drag coefficients of $C_D = 3.9$ were found for live blue mussel lines and $C_D = 3.4$ for the surrogate model, for conditions of Keulegan–Carpenter number ($KC$) 10 to 380, using regular wave tests.

**Keywords:** aquaculture; bivalves; waves; physical model tests; Large Wave Flume

## 1. Introduction

The world's population is projected to increase to ten billion people by the middle of the twenty-first century [1]. Humankind faces the enormous challenge to establish sustainable food production structures as described in the seventeen Sustainable Development Goals in the United Nations' 2030 Agenda [2]. One aspect to a multi-faceted solution could be an efficient and sustainable marine aquaculture bivalve production [3,4], with all its health benefits as prevention of cardiovascular diseases or age-related macular degeneration [5,6]. The total aquaculture production is predicted to reach 109 million tons in 2030, as reported by the [7]. In contrast to freshwater aquaculture or agriculture, marine aquaculture has little to no dependence on the scarce resource freshwater [8,9] or on available arable land [10,11]. Capture production is stagnating at 95 million tons per year [7,12] due to overexploitation and resulting legal restrictions [13,14]. However, in 2018, only 10% of total aquaculture production was molluscs [7]; although fish and crustaceans marine aquaculture depends on feeds from wild fisheries [15], approximately 20 to 25 million tons of fish meal is required to produce only 30 million tons of fish and

crustaceans [16], and lowering the trophic level of food production is commonly suggested to lower environmental impacts [15,17–19]. Still, even lower intensity aquaculture is prone to escaping individuals, which could become invasive or genetically alternative stocks, distribution of parasites and diseases, release of antibiotics and drugs, eutrophication due to marine litter, loss of benthic biodiversity, change of local hydrodynamics, change of species assemblage including mammals and reduction of the native stocks by spat recruiting [17,20–22]. More specific to bivalve marine aquaculture, filter feeders alter the composition of the water column, leading to changes in bloom, light penetration and primary production, which could be positive or negative, depending on the biological situation [23]. Sustainable technologies, integrated farm siting, effluent management, disease control, and culture of native species, as well as government, market and self-regulation, are suggested to overcome these impacts [24]. In addition, rising space limitations near shore, as well as conflicts between food supply, infrastructure and tourism [15,18,25], are drivers to relocate to offshore locations [25–27], which would also reduce negative environmental impacts and increase carrying and assimilative capacity [28]. Reference [29] describes offshore aquaculture developments in Belgium, France, Germany, Ireland, Italy, Netherlands, New Zealand, the UK and the USA, of which multifunctional systems are researched by [3] in wind farms in the North Sea or [30] for oil and gas platforms in the Gulf of Mexico. However, the more energetic wind, wave and current environment offshore requires research on reliable offshore marine aquaculture structures, as well as harvest and monitoring technology [4,25,31,32].

Bivalve marine aquaculture production can be classified into on-bottom and off-bottom, or suspended culture [33], which is divided in intertidal, raft or longline systems [34]. In intertidal systems, mussels grown on collecting ropes or in net sockings, which are wound up around wooden poles after spat collecting offshore [29,35]. In raft systems, mussel ropes are hung from swimming or submerged platforms [36]. Longline systems, found to show the highest crop yield by [35], consist of two parallel backbone ropes supported by buoys from which mussel ropes, so called droppers, are hung in loops down to ten to fifteen meters in depth. The system is moored with single warps to anchor blocks [37]. Reference [29] emphasizes that off-bottom culture, in general, makes better use of the water column and is less vulnerable to predators, and that longline systems are cheaper than rafts, are easily constructed and maintained, are more suitable to winter storm conditions, and allow a highly mechanized culture, which is why [4] suggest that bivalve longline culture should be researched for offshore applications. However, even research on nearshore bivalve structures is scarce, a fact that makes accurate design assumptions and design force estimation difficult.

The effect of mussel farms on the surrounding current and wave conditions was previously investigated [37–39]; the first observations of a longline farm in the field were conducted [40,41], and numerical models for whole longline farms were set up [42]. What is widely missing, as essential parameters for the design of offshore longline structures, are experimental based force coefficients on mussel dropper lines [42,43]. These coefficients were so far only presented for rigid oyster trays [44], suspended canopies [45], biofouled nets [46] or bivalve encrusted piles [47–49]. Drag and inertia coefficients of smooth, rough or flexible cylinders and combinations of those are more widely researched. Reference [50] laid the foundation with the MOJS equation (Equation (4)) for the force exerted on piles by currents or waves. Reference [51] compared several methods for the determination of force coefficients of heavily roughened cylinders from experimental force data and recommend the Least Square Method, as well as wave-by-wave comparison with mean drag coefficient ($C_D$) 1.88 and mean inertia coefficient ($C_M$) 2.08 for a pile with diameter of 5.13 cm. Reference [52] found that the same method can be applied, to a certain extent, to smooth flexible cylinders in random waves when the relative motion leads to wide scatter or negative values for the coefficients. Using the Least Square Method and relative motion analysis mean drag coefficient at the middle of the cylinder is 1.9 and mean inertia coefficient is 0.87. To the authors' knowledge, reference [37] were the first to estimate

the drag coefficient of dropper lines as 0.89 with a towing test, but later chose values of reference [51] for their mussel farm model because that value was difficult to apply in waves. Reference [53] were the first to conduct extensive towing and wave tests with live blue mussel dropper lines for experimentally verified force coefficients for offshore bivalve farm design. In light of the large difficulties with live shellfish in freshwater-based laboratory wave tanks, the quality of a dropper line surrogate for wave tests [54], as a strategy of substituting live shellfish testing, was additionally yielding a mean drag coefficient of $C_D$ = 2.3 for the live line and $C_D$ = 2.4 for the surrogate, as well as mean inertia coefficient of $C_M$ = 2.1 for the live line and $C_M$ = 2.3 of the surrogate.

Based on the above presented state-of-the-art review on mussel marine aquaculture and force determination of bivalve-encrusted aquaculture gear, the overall objective of this work is to enhance the knowledge on the force regime acting on ultra-rough, bivalve covered surfaces and ropes, which is still strongly debated and more accurate design basis could be obtained. Despite the overall force acting on shellfish-encrusted rope, that is, a result of global velocity and pressure field conditions surrounding the immersed specimen, it has often been an aspect of research how strong individual mussel specimen adhere to a piece of rope. This aspect is important, and yet unaddressed in large-scale experimental settings, as soon as the local drag and pressure-related forces exceed the ability of the animals to hold on to the rope. Hence, the following specific objectives are addressed to shed light on the above knowledge gaps:

- To understand the adhesive forces of live mussels attached to farming rope;
- To quantify the drag and inertia coefficients of bivalve-encrusted dropper lines when exposed to waves in large, near full-scale experiments;
- To comparing the force response of live mussels with a surrogate specimen.

To the authors knowledge, this is the first near full-scale experimental series with live mussel dropper lines in waves and only the second comprehensive experimental campaign with live mussel dropper lines [53].

## 2. Materials and Methods

### 2.1. Testing Specimen

The experiments are based on ten marketable nine-months-old juvenile dropper lines grown with blue mussels (*Mytilus edulis*) from an aquaculture farm at the Baltic Sea, at the Bay of Kiel, Germany, whereof six selected lines were tested in the Large Wave Flume. Blue mussels were selected as cultured species due to their high importance in aquaculture: They made up for 82% of global mussel fisheries in 2011, due to the high protein content [29]. At the local scale, blue mussels are widely distributed from sheltered to high wave-exposed conditions, from marine to estuarine regions, from subtidal to intertidal shores and various substrates as wood, rock, cement or shell [29]. That extensive distribution pattern displays their euryhaline and eurythermal capabilities [34,55–61], as well as their independence of respiration and filtration over several hours [61].

For attachment, mussels secret a byssus containing several threads off their foot [62,63] where thread thickness can be increased to adapt to changes in water motion [64,65]. Mean maximum dislodgement forces for blue mussels grown on polyester nets formed as tubes was 3.6 N and grown on artificial seaweed 5.6 N [66]. A peak force of 15 N was measured for a single blue mussel attached by nine byssus threads by [67]. Reference [68] found mean attachment strength of 0.6 N for blue mussels of 1 to 3 cm off the North Sea and [69] found 2.5 N for blue mussels of 6.8 to 8.8 cm length.

Reference [29] lists the following environmental factors for mussel growth as the most important modulators: temperature, salinity, water flow, water depth, tidal level, wave action, pollutants, aerial exposure and stock density, as well as endogenous factors, such as genotype and physiological status. Reference [70] shows that, at the Baltic Sea, mussels grew to 4 cm length at 4 m depth and to 2.5 cm length in 15 m depth in 12 years. Reference [71] reports mussel lengths between 2.5 and 6.5 cm for mussels of the East Yorkshire Coast aged five to ten years old. Mussels harvested on several materials in the

Jade estuary reached a mean length of 4.61 cm, mean width of 5.08 cm and mean height of 1.67 cm after 16 months [66]. Mean length and mean width of mussels grown on a dropper line at the Baltic Sea near Kiel are 4.7 and 2.2 cm, respectively [54].

### 2.2. Surrogate Model of the Live Mussel Dropper Lines

Reference [54] gives a detailed description of the creation of three mussel dropper lines surrogates based on a 3D-Scan of an adult blue mussel line encrusted with newly seeded spat from the same aquaculture farm as the lines tested in this work. The authors used the surface descriptor Abbott–Firestone Curve [72] to develop surrogate model geometries with similar characteristics regarding the weighed arithmetic average material distribution. The best fit regarding drag and inertia coefficients in towing and wave tests, with towing velocities from 0.25 to 1.00 m/s and wave height from 0.1 to 0.15 m, with wave period 1.2 to 2.4 s, respectively, is a surrogate created by adding uniform mussels to a slender cylinder at different angles of incidence until mean weight was equal to the original mussel data [53]. An identical, appropriately scaled surrogate model is used in this work; the mussel dropper line is made by threading twenty individual surrogate sections of 8.42 cm height onto a 2.2 m wire rope, corresponding to the total length of the live dropper lines. The characteristics of the surrogate are listed in Table 3 for comparability to the live dropper lines.

### 2.3. Experimental Setup

The experiments are conducted in the Large Wave Flume of the Coastal Research Center in Hannover, Germany; a joint facility of the Technische Universität Braunschweig and the Leibniz Universität Hannover. The Large Wave Flume has an effective length of 307 m, a width of 5 m and a depth of 7 m. The hydraulically driven wave machine (900 kW) gives a maximum stroke of 4 m to the wave paddle, generating waves up to a height of 2 m, allowing for quasi-prototype conditions of regular, irregular and breaking waves to be tested. Wave generation is controlled by an active absorption system. Figure 1 depicts a side view of the Large Wave Flume, including the wave paddle (x = zero, reference position), measuring equipment, test rack with mussel dropper lines and wave dampening beach. Forward wave propagation direction is defined as x-direction; vertical direction is defined as z-direction.

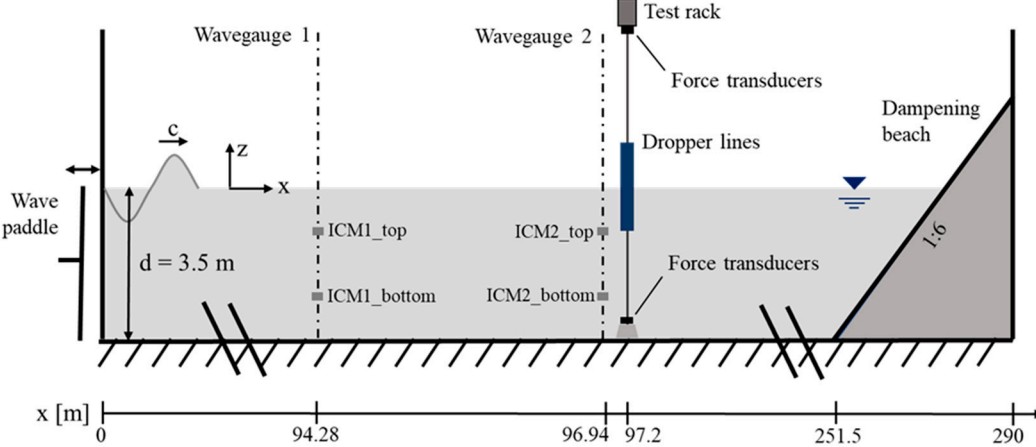

**Figure 1.** Side view of the Large Wave Flume with wave paddle, measuring equipment, test rack with mussel dropper lines and wave dampening beach (sketch not to scale).

The test rack (Figure 2) is located at a distance of 97.2 m to the wave paddle. The test rack includes a trapezoidal concrete slab at the bottom of the flume, a t-beam spanning the width of the flume as well as three mussel dropper lines clamped between the concrete slab and the t-beam with wire ropes, allowing for free rotation, but restricting natural motion.

This setup does not depict natural behavior of suspended dropper lines but allows an efficient determination of force coefficients. The mussel dropper lines have a length of 2 m and are positioned in a distance of 1.5 m to the flume walls and 1.0 m to each other. The upper wire ropes, connecting the t-beam and the mussel lines, include a rope tensioner to set uniform pretension conditions for all mussel lines. The lines are submerged by half-length, to prevent wave acting on the wire ropes at all times, as well as to allow for observation throughout all runs.

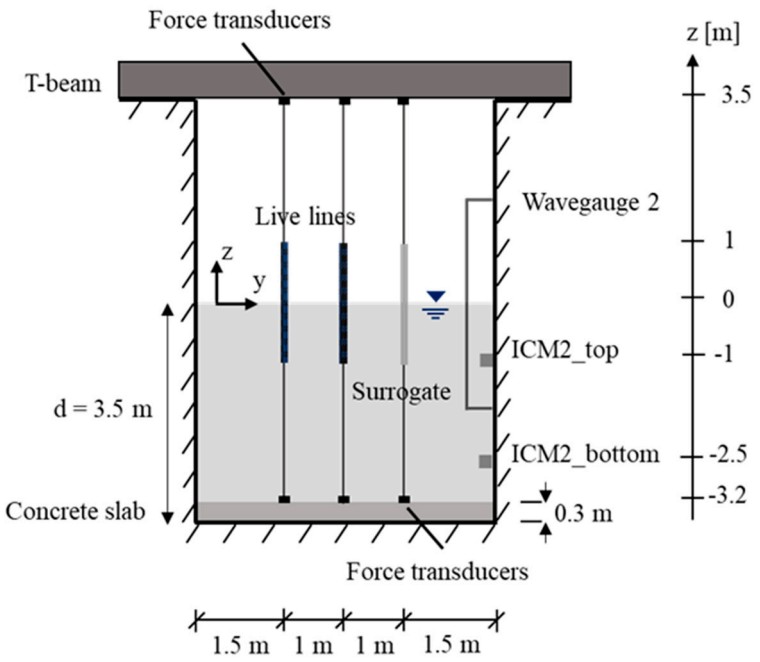

**Figure 2.** Test rack with live dropper lines, surrogate and measuring equipment (sketch not to scale).

*2.4. Instrumentation*

Wave induced forces in x- and y-direction on each of the dropper lines are measured with load cells (CTL100, XSENSOR, Darmstadt, Germany) at the top and bottom of each line. The CTL100 has a nominal load range of up to 100 kg, a combined error of 0.02 kg and is hermetically sealed according to IP68 standard. The inline tension is recorded by one-axis force transducers (U2B, HBM, Darmstadt, Germany). The U2B has a nominal force range of 2 kN, a combined error of 2 N and IP67 specification. The sampling rate for both force sensor groups is set to 1000 Hz. An identical instrumentation setup is used for all three line locations, as shown in Figure 2.

Incident wave conditions are measured in close vicinity to the dropper lines by two wave gauges and four inductive current meters. All wave gauges are attached to the South wall of the flume. The wave gauges are wire type wave gauges based on combined electrical resistance and capacity technique and consist of two electrodes, a measuring wire and a ground plane, with an accuracy of ±1 mm. The sampling rate is set to 100 Hz. Wave gauge 1 is positioned in a distance of 94.28 m to the idle position of the wave paddle, and wave gauge 2 is positioned in a distance of 96.94 m corresponding to a distance of 2.92 m and 0.26 m to the test rack, respectively. Right next to these two wave gauges, two inductive current meters (ISM-2001F, hs engineers, Hannover, Germany) are positioned each in different heights so that water particle velocity u and v in x- and z-direction is acquired at four spots, twice in a distance of 2.92 m (ICMs1) and twice in a distance of 0.26 m (ICMs2) to the test rack as well as 0.31 m to the wall of the flume. Velocity in y-direction is assumed to be negligible which is proven by evaluating measured force in y-direction. The lower current meters (ICMs_bottom) are located at a height of 1 m above the flumes bottom each, while the upper current meters (ICMs_top) are attached in a height

of 2.5 m, corresponding to 2.5 and 1 m below water surface, respectively. The current meters have a nominal velocity range of ±3 m/s and a measuring accuracy of ± (0.5% reading + 0.5% limit range). The sampling rate is set to 100 Hz. All testing is recorded by a GoPro Hero4 with a high-definition resolution and a sample rate of 100 fps.

*2.5. Experimental Procedures*

Ten nine-month-old blue mussel dropper lines of 2.5 m length are transported and stored in a water tank filled with Baltic Sea water from the Bay of Kiel. Temperature and aeration are controlled throughout transport and storage prior to the experiments. The mussels are fed with approximately 4 g of plankton every evening. Each dropper line is cut into 2.1 m length, weighed and numbered for later distinction (labeled from Nos. 1 to 10). The width of each line is measured in approximately 0.1 m steps, to determine mean diameter and its standard deviation. Volume determination is conducted by measuring their displacement in a water-filled container of known dimensions. In the following, density, weight per meter, ratio of dropper weight to clamp weight and equivalent diameter is determined. The equivalent diameter, $D_{eq}$, is calculated as the diameter of an ideal cylinder with the same volume, $V$, as the mussel dropper line of length, $L$.

$$D_{eq} = \sqrt{4 \cdot \frac{V}{\pi \cdot L}} \tag{1}$$

The live mussel dropper lines Nos. 1 to 6 are clamped (Figure 3a) and mounted into the test rack, with remaining 2 length, so that two live lines and the surrogate were tested at the same time. The lines are exposed to thirteen wave trains with thirty single waves of targeted wave heights $H$ of 0.2 to 1 m and targeted wave period $T$ of 1.5 to 8 s (Table 1 and Figure 4). Still water depth is 3.5 m, so that each dropper is submerged over a length of 1 m. In between all wave tests, a waiting period ensures settling of the water level, to avoid biased influence of previous tests. In addition, rack-only tests are conducted for each wave train, to obtain forces acting on the blue mussel dropper line alone, without forces acting on the test rack.

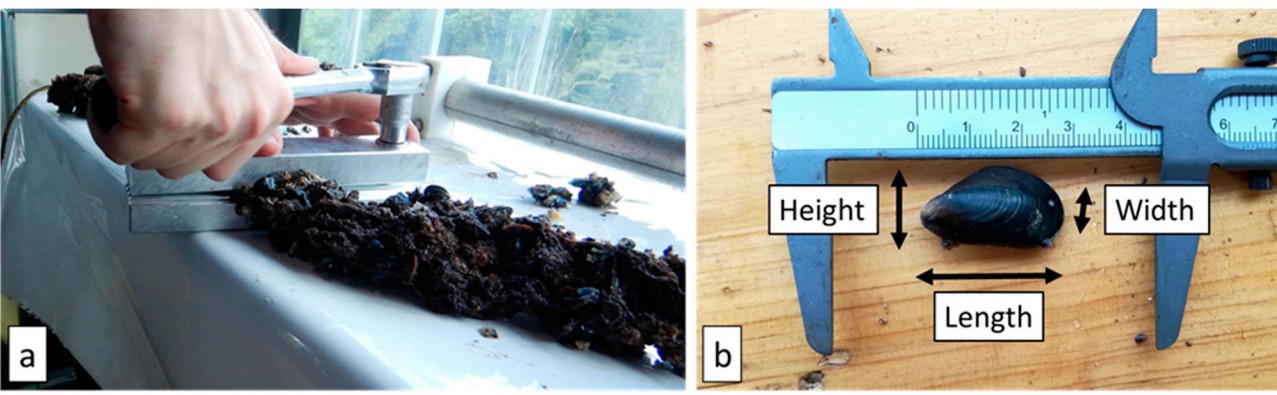

**Figure 3.** (**a**) Clamping of a live line for setup in the test rack. (**b**) Measurement of an individual mussel regarding the common conventions for mussel length, height and width.

**Table 1.** Wave parameters for experiments at a water depth of 3.5 m.

| Wave Train | 1 | 2 | 3 | 4 | 5 | 6 | 7 | 8 | 9 | 10 | 11 | 12 | 13 |
|---|---|---|---|---|---|---|---|---|---|---|---|---|---|
| Height H (m) | 0.2 | 0.2 | 0.4 | 0.4 | 0.4 | 0.6 | 0.6 | 0.6 | 0.8 | 0.8 | 0.8 | 1.0 | 1.0 |
| Period T (s) | 2.0 | 2.5 | 3.0 | 3.5 | 4.0 | 4.5 | 5.0 | 5.5 | 6.0 | 6.5 | 7.0 | 7.5 | 8.0 |
| Length L (m) | 6.2 | 9.6 | 13.1 | 16.6 | 20.0 | 23.3 | 26.5 | 29.7 | 32.9 | 36.0 | 39.1 | 42.1 | 45.2 |
| Steepness H/L | 0.03 | 0.02 | 0.03 | 0.02 | 0.02 | 0.03 | 0.02 | 0.02 | 0.02 | 0.02 | 0.02 | 0.02 | 0.02 |

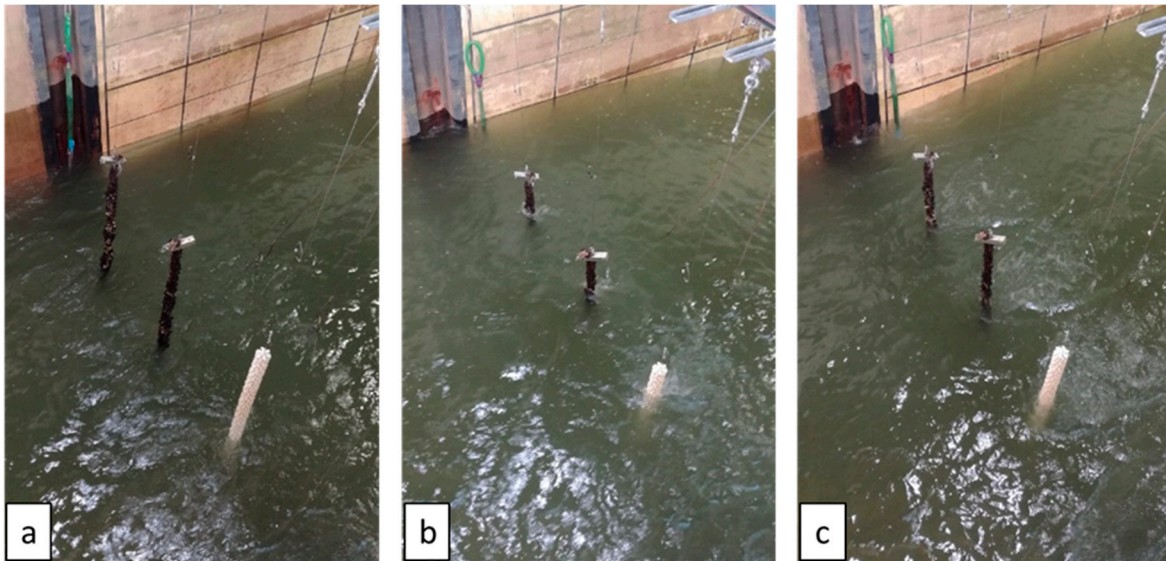

**Figure 4.** Exposure of dropper lines in waves. (**a**) Forward motion in wave trough. (**b**) Backward motion in wave crest. (**c**) Wake formation behind lines at approximately zero crossing.

The remaining parts of the original blue mussel lines (cutoffs) are used in pull-off tests, to determine maximum attachment strength of single mussels, to help and provide new data on the tested shellfish individuals under experimental conditions in large-scale testing. Adhesive forces are measured with a FMI-100B5, by Alluris, with nominal range 50 N, measuring accuracy ± 0.2% at 23 °C, resolution 0.01 N and sampling rate 1000 Hz. Figure 5 shows the procedure: the device pulling at a single mussel by backward motion on a carriage, controlled by a rotary screw for repeatability. The display of the force gauge is recorded with a Nikon Coolpix L810 with a sampling rate of 25 Hz. The recordings are evaluated, frame by frame, for time histories of adhesive force, as well as maximum force. Individual mussels that were pulled off are additionally selected for data acquisition of length, height and width (Figure 3b). The sample is evaluated with descriptive statistics parameters. Outliers are defined as values which are more than 1.5 times of the interquartile range above the third quartile or below the first quartile and are removed from the dataset, to obtain the statistic parameters.

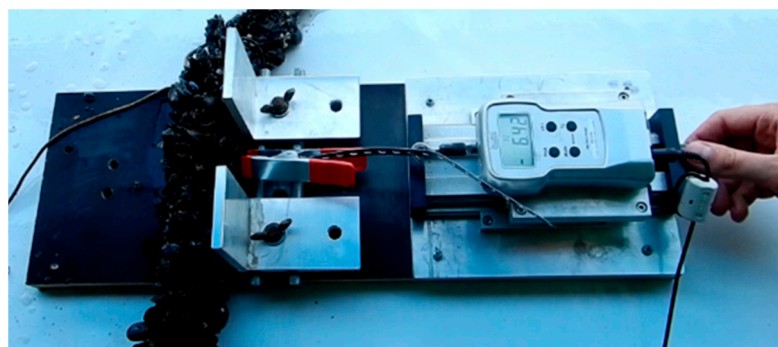

**Figure 5.** The attachment strength of an individual mussel to a line cutoff is measured with a force transducer.

The data of the wave gauges, current meters and load cells are evaluated regarding water-level evaluation, orbital velocity and wave-induced forces on mussel lines. Water-level elevation of wave gauge 1, as well as horizontal and vertical particle velocity at ICM2, is only used to show that there is neither change in wave height nor particle velocity over the distance of 2.66 m from wave gauge 1 to wave gauge 2 or ICM1 to ICM2. Hence, water

elevation of wave gauge 2 and horizontal particle velocity of ICM1 are used to describe the wave conditions at the test rack after applying the respective time shift. Inline tension is used for attaining equal pretension of all dropper lines in the test setup while the measured force in y-direction is used to state 2D conditions in the Large Wave Flume.

The raw data is filtered with an eight-order, low pass Butterworth filter with sampling frequency of 100 Hz and cutoff frequency of 5 Hz. The cutoff frequency is determined by evaluating Fast-Fourier Transformations of each raw data sets for maximum amplitude spectra. The frequency dependent phase shift due to the Butterworth filter is corrected. The filtered time histories are cut to the relevant time frames, leaving 28 to 6 waves from wave train 1 to wave train 13 for further evaluation. Every single wave of each wave train is analyzed regarding wave height and wave period. Table 2 lists exemplarily the results for wave height and period at wave gauge 2 for the tests with line 1, line 2 and the surrogate as well as averaged results over all tests with mean and standard deviation (std).

**Table 2.** Wave height and period of individual waves at wave gauge 2.

| Wave Train | No. of Waves | Tests with Line 1, Line 2 and Surrogate | | | | All Tests | | | |
|---|---|---|---|---|---|---|---|---|---|
| | | Mean $H$ (m) | Std $H$ (m) | Mean $P$ (s) | Std $P$ (s) | Mean $H$ (m) | Std $H$ (m) | Mean $P$ (s) | Std $P$ (s) |
| 1 | 28 | 0.19 | 0.01 | 2.00 | 0.03 | 0.19 | 0.01 | 2.00 | 0.04 |
| 2 | 28 | 0.18 | 0.00 | 2.50 | 0.04 | 0.19 | 0.01 | 2.50 | 0.04 |
| 3 | 23 | 0.39 | 0.01 | 2.99 | 0.06 | 0.39 | 0.01 | 3.00 | 0.06 |
| 4 | 18 | 0.42 | 0.01 | 3.49 | 0.06 | 0.42 | 0.01 | 3.50 | 0.05 |
| 5 | 15 | 0.43 | 0.01 | 3.98 | 0.05 | 0.43 | 0.01 | 4.00 | 0.06 |
| 6 | 13 | 0.64 | 0.01 | 4.51 | 0.06 | 0.64 | 0.01 | 4.50 | 0.06 |
| 7 | 11 | 0.63 | 0.01 | 4.98 | 0.05 | 0.64 | 0.01 | 5.01 | 0.05 |
| 8 | 10 | 0.64 | 0.00 | 5.49 | 0.06 | 0.64 | 0.01 | 5.50 | 0.05 |
| 9 | 8 | 0.83 | 0.00 | 5.98 | 0.05 | 0.84 | 0.01 | 5.99 | 0.05 |
| 10 | 8 | 0.85 | 0.01 | 6.53 | 0.06 | 0.85 | 0.01 | 6.53 | 0.07 |
| 11 | 7 | 0.84 | 0.01 | 7.02 | 0.05 | 0.84 | 0.02 | 7.02 | 0.06 |
| 12 | 6 | 1.09 | 0.01 | 7.51 | 0.02 | 1.10 | 0.01 | 7.52 | 0.04 |
| 13 | 6 | 1.06 | 0.01 | 8.03 | 0.08 | 1.07 | 0.02 | 8.02 | 0.06 |

The force time histories used for the investigation are corrected to obtain forces acting on the mussel dropper line alone, without forces acting on the test rack, by subtracting the filtered and cut time histories of the measured forces of the rack-only tests. Comparability of the wave conditions generating these forces is proven by comparison, firstly, of the mean minima and maxima of each wave train as well as their phase shifts in the water level elevation and, secondly, of the wave height and period of each single wave. Since the rack-only tests are conducted without the steel clamps, the correction of the measured forces results in the hydrodynamic forces on the mussel dropper lines plus steel clamps. Both force time histories in x-direction, measured at the top and the bottom of the test rack, are added to a total force history in x-direction on each line. The quality of the surrogate is evaluated by comparing mean force peaks of each wave train $i$ between live lines $F_{P,Line}$ and surrogate $F_{P,Surrogate}$ of the joint test run, firstly, with the normalized mean error.

$$MNE = \frac{100}{N_{Wt}} \cdot \sum_{i=1}^{N_{Wt}} \frac{F_{P,Surrogate,i} - F_{P,Line,\ i}}{F_{P,Surrogate,i}} \tag{2}$$

with $N_{Wt}$ as total number of wave trains acting on the lines and, secondly, with the root mean square error.

$$RMSE = 100 \cdot \sqrt{\frac{1}{N_{Wt}} \cdot \sum_{i=1}^{N_{Wt}} \left[ \frac{F_{P,Surrogate,i} - F_{P,Line,\ i}}{F_{P,Surrogate,i}} \right]^2} \tag{3}$$

In addition, each force peak of the live lines is compared to the respective force peak of the surrogate of the joint rest run. The data of the three test runs of the surrogate is also compared, regarding the force peaks, using mean normal error (MNE) and root mean square error (*RMSE*).

The time histories of summed force in x-direction is applied in the MOJS equation of [50] to estimate drag and inertia coefficients of the mussel dropper lines. The MOJS equation superimposes drag force, $F_D$, and inertia force, $F_M$, to a line force, $F$, in horizontal direction on an object in surface waves, using horizontal particle velocity, $u$, and acceleration, $du/dt$:

$$F(t,z) = F_D(t,z) + F_M(t,z) = C_D \cdot \rho_W \cdot \frac{D}{2} \cdot u(t,z) \cdot |u(t,z)| + C_M \cdot \rho_W \cdot \pi \cdot \frac{D^2}{4} \cdot \frac{du(t,z)}{dt} \quad (4)$$

where time is $t$, immersion depth is $z$, drag coefficient is $C_D$, inertia coefficient is $C_M$, water density is $\rho_W$ and object diameter is $D$. Integration of the line force, $F(t,z)$, over the entire immersion depth gives $F_{tot,t}$ as approximated total force in x-direction on the object at time step, $t$. The MOJS equation was already successfully used on live mussel lines in related research activities [53,54,73]. Water density is set to 1000 kg/m3, the freshwater value. As the equivalent diameter, $D_{eq}$, object diameter, $D$, is used. Since water particle velocity was not measured along the entire immersion depth of the dropper lines, the required horizontal particle velocity $u(t,z)$ and acceleration $du(t,z)/dt$ is calculated with Stokes 3 Wave Theory, yielding the smallest mean square error for the measured velocities of the ICMs. The theoretical maximum horizontal particle velocity umax at z = 0 m is additionally used to determine the Keulegan–Carpenter number (*KC*, [74] of each single wave of each wave train. The *KC* number is calculated as the ratio of drag force and inertia force, as follows:

$$KC = \frac{u_{max} \cdot T}{D_{eq}} \quad (5)$$

with the individual wave period as $T$ and the equivalent diameter of the dropper line as $D_{eq}$. Combinations of drag and inertia coefficients $C_D$ and $C_M$ are applied to the MOJS equation to calculate theoretical line force, $F$, in the horizontal direction, on the mussel dropper line for each single wave of each wave train [51]. The line forces over the whole immersion depth are integrated to attain the approximated force, $F_{app,t}$, for each time step. Total force on the mussel lines is measured, so the most fitting combination of force coefficients $C_D$ and $C_M$ can be estimated by minimizing the least square error, $\varepsilon^2$, between measured force, $F_{meas,t}$, and MOJS approximated force, $F_{app,t}$, with the function following function:

$$\varepsilon^2(C_D, C_M) = \int_0^T |F_{meas,t}|^k \cdot [F_{meas,t} - F_{app,t}(C_D, C_M)]^2 dt \quad (6)$$

where the weighting factor is $k$ [51,54]. In this evaluation, $k$ is set to zero, so that all deviations are weighted equally [75]. The best fit of $C_D$ is depicted with corresponding *KC* number. Since the least square error addresses the derivation over the whole time history of one individual wave, additionally, the root mean square error RMSE (Equation (3)) between the force peaks $i$ of time histories $F_{meas}$ and $F_{app}$ is determined with the total number of evaluated force peaks in $N$. Last, mean values, as well as standard deviation of force coefficients, are calculated for the lie lines and the surrogate.

## 3. Results

### 3.1. Dropper Line Testing

The results of specimen analysis, that is weighting and measuring the ten nine-month-old mussel dropper lines before wave testing, are summarized in Table 3. The density, weight per meter and the ratio of the weight of the line to the weight of the clamps range widely, while the mean diameter spans from 7.5 to 10.3 cm. The equivalent diameter, depending on the individual volume, again ranges widely from 2.99 to 6.72 cm, whereas [54]

determined a mean equivalent diameter of 10.31 cm for their more mature dropper lines from the same shellfish farm in the Bay of Kiel. The equivalent parameters of the surrogate are added in Table 3, to facilitate comparisons. Compared to lines 1 to 6, used in the wave tests, the surrogate is only fifth in regard to total weight, but second regarding mean and equivalent diameter. Overall, due to the biological variabilities, scattered results for the force coefficients of the live mussel lines are expected, and less variance for the force coefficients of the surrogate is expected.

**Table 3.** Results of the dropper line testing.

| Line No. | Density (kg/m$^3$) | Weight (kg/m) | Total Weight with Clamps (kg) | Ratio Weight Line to Weight Clamps | Mean Diameter (cm) | Std Diameter (cm) | Equivalent Diameter (cm) |
|---|---|---|---|---|---|---|---|
| 1 | 2593.55 | 1.82 | 6.53 | 0.70 | 8.09 | 1.79 | 2.99 |
| 2 | 1757.76 | 1.87 | 6.63 | 0.68 | 8.80 | 2.33 | 3.69 |
| 3 | 2841.60 | 3.86 | 10.79 | 0.33 | 10.31 | 2.80 | 4.16 |
| 4 | 1389.37 | 2.89 | 8.76 | 0.44 | 9.44 | 1.56 | 5.14 |
| 5 | 1304.05 | 4.63 | 12.41 | 0.28 | 9.66 | 1.87 | 6.72 |
| 6 | 1241.59 | 3.62 | 10.30 | 0.35 | 9.13 | 1.86 | 6.09 |
| 7 | 1797.95 | 2.13 | No wave test | No wave test | 8.36 | 1.33 | 3.99 |
| 8 | 363.58 | 0.85 | No wave tests | No wave tests | 7.79 | 1.63 | 5.47 |
| 9 | 922.39 | 1.19 | No wave tests | No wave tests | 7.52 | 1.67 | 4.05 |
| 10 | 509.21 | 1.02 | No wave tests | No wave tests | 8.04 | 1.52 | 5.04 |
| Sur. | 1220.00 | 4.25 | 8.5, no clamps | No clamps | 10.30 | - | 6.62 |

During the experiments, drop-off of individual mussels and groups of mussels is observed. Minor loss is observed after wave train 9, and strong loss is observed after wave train 11. After wave train 13, approximately two-thirds of mussels are lost. A photo documentation of the mussel drop-off process is depicted in Figure 6, with a sequential loss with increasing experimental time. Exemplarily, four mussel dropper lines were weighted after exposure to all wave trains. Minimum drop-off weight was 71.4% and maximum 88.1%, resulting in a mean weight loss of 67% for this small sample, throughout thirteen wave trains with thirty waves. Since only equivalent diameter before testing could be applied in the evaluation, an overestimation of the force coefficients might be present for high *KC* numbers. The mussel drop-off is hypothesized to occur as the blue mussels may have got into a state of stress through the increase of water temperature, external force increase and freshwater conditions.

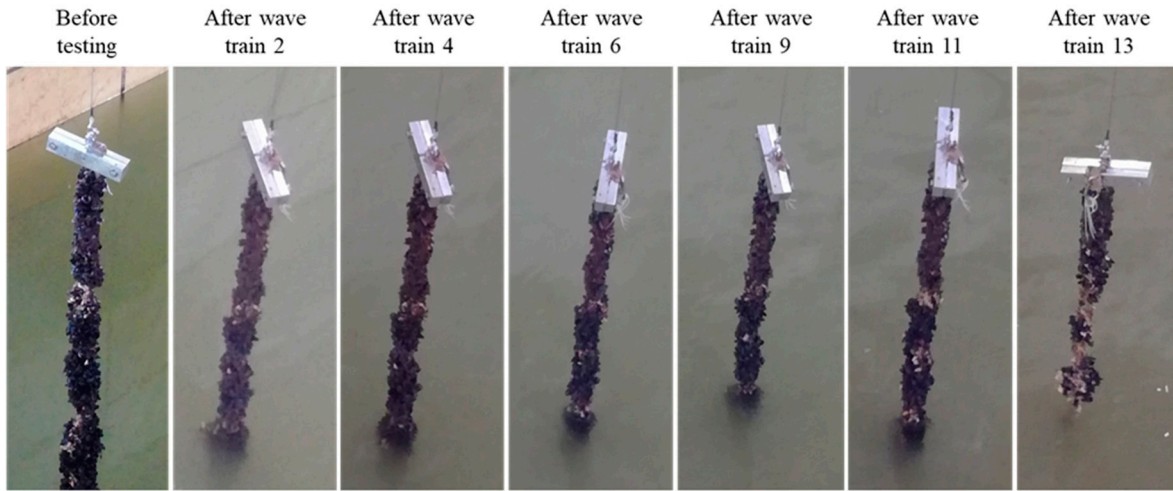

**Figure 6.** Mussel drop-off for line 6, depicted from before the tests to after the last test, resulting in a loss of weight of 88.1%.

### 3.2. Individual Mussel Attachment Strength

A set of 82 pull-off tests was successfully conducted. Figure 7 depicts a box plot for each measurement in this sample set, as well as a histogram of attachment strength that follows a Gaussian distribution. One outlier was found and is marked with a red cross. Maximum adhesive force is 10.55 N, while minimum adhesive force is 0.48 N. The mean of the measurements is $5.07 \pm 2.26$ N. References [76,77] suggest connecting attachment strength with shell area, which was not measured in this work. Comparing the results with mussels grown on polyester nets formed as tubes of the work of [66], the mean adhesive force measured found in this work is higher. This is indicating a well-chosen substrate for mussel farming for our live mussel dropper lines, as material is crucial for settlement [66]. The peak force of 15 N measured by [67] is reached by one mussel with 12.69 N, but is declared as an outlier, and the sample quantity of [67] only contained one single test.

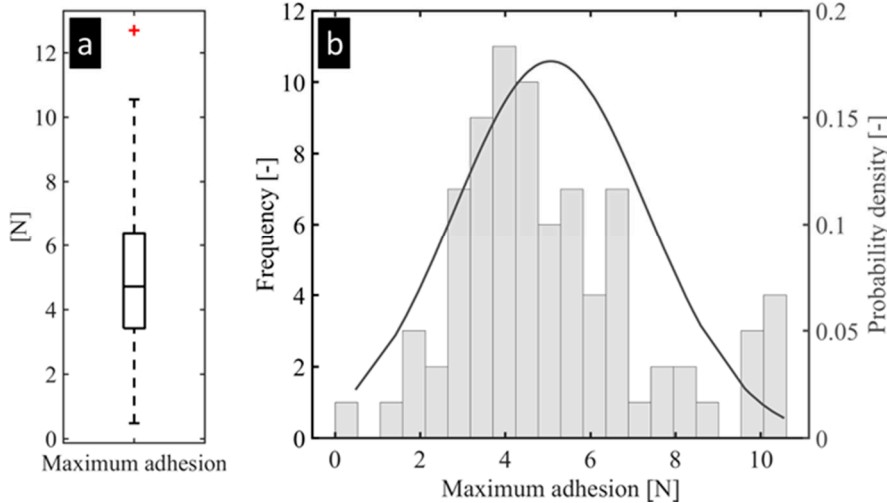

**Figure 7.** (**a**) Box plot, outlier as red crosses, and (**b**) histogram and probability distribution for maximum attachment strength of individual mussels in a sample of 82 mussels.

The evaluation of the camera recordings of the pull-off tests showed varying failure mechanisms. Three classes are suggested to cluster the observed failure mechanisms more methodologically: slow, instant and mixed fracture. Examples of equivalent measured force time histories are depicted in Figure 8. A slow failure mechanism is characterized as slow drop to zero after measured force is reaching its maximum, interpreted as the yield strength of the byssus material. It is assumed that only single byssus threads or small groups thereof tear at a time while the remaining, intact threads continue to withstand. In contrast to that, an immediate force drop towards zero is defined as instant fracture. Presumably, all threads tear quasi-simultaneously after reaching the maximum adhesion force by yielding. In the depicted cases in Figure 8, measured force drops to zero within 0.5 s for the instant failure and within 12 s for the slow break. The last class is a combination of the slow and instant fracture, where either a fast force drop is followed by a slow decrease to zero or vice versa. In the 82 pull-off tests, 39 slow, 26 instant and 17 mixed failures were observed. The results of the camera recordings go in line with previous results of [65], who describes the yielding of threads as the essential ability to recruit threads of different length for higher attachment strength.

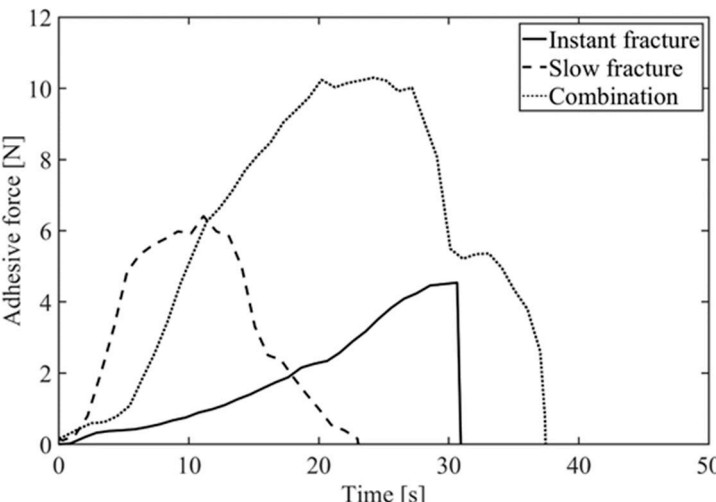

**Figure 8.** Force time histories of the adhesive force prior to pull-off as examples of classification of failure of mussel attachment.

### 3.3. Properties of Mussel Individuals

A sample of 102 single mussels was used for determining mussel properties. The mean values for the nine-month-old juvenile mussels correspond to a mean single mussel length of 3.04 ± 0.36 cm, a mean height of 1.69 ± 0.16 cm and a mean width of 1.25 ± 0.18 cm. In more detail, Figure 9 depicts a box plot for each measured single mussel parameter in this sample. Outliers are marked with red crosses. Individual mussel length has the highest sample standard deviation, which is approximately double the values for height and width. Comparing the results to the literature, we see that the mean mussel parameters are lower than the values of [70], with mussels at the age of twelve years off the Baltic Sea at 4 m depth, but fit into the range of [71], who characterized dimensions of mussels at the age of five to ten years, off the East Yorkshire Coast. Although these indications from the literature refer to samples four-to-nine-years older, the wave impact at their settlement location, a rocky shore, could be the reason for slower growth [77], whereas the Bay of Kiel had favorable conditions, with open water and fairly little wave-energy exposure. A good comparison is the sixteen-month-old mussels in the tests of [66], off the Jade estuary, that were seven months older and a few decimeters longer, higher and wider, indicating similar growing conditions. The mussels grown on the dropper lines from the same farm in Kiel are found to be slightly larger and wider [54]; therefore, they are older, assuming that the farming site was chosen due to constant growing conditions.

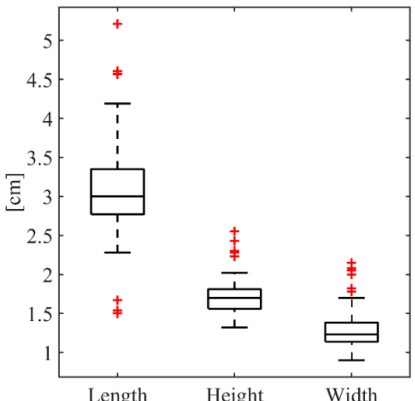

**Figure 9.** Box plot for length, width and height of individual mussels in a sample of 82 mussels; outliers are marked as red crosses.

### 3.4. Force Time Histories

Example force time histories of wave-induced forces in x-direction on the dropper lines tested in the Large Wave Flume are displayed in Figure 10 for wave trains 2, 5, 9 and 12. Provisional visual inspection of the data and evaluation allows firstly to state that similar forms of time histories are observed for all lines and all wave trains. For smaller wave trains, the surrogate shows higher force peaks than all live lines. Starting with wave train 4, the surrogate underestimates the force peaks of the live lines. Generally, the dynamics of the immersed shellfish dropper lines remain constant throughout the experimental conditions tested.

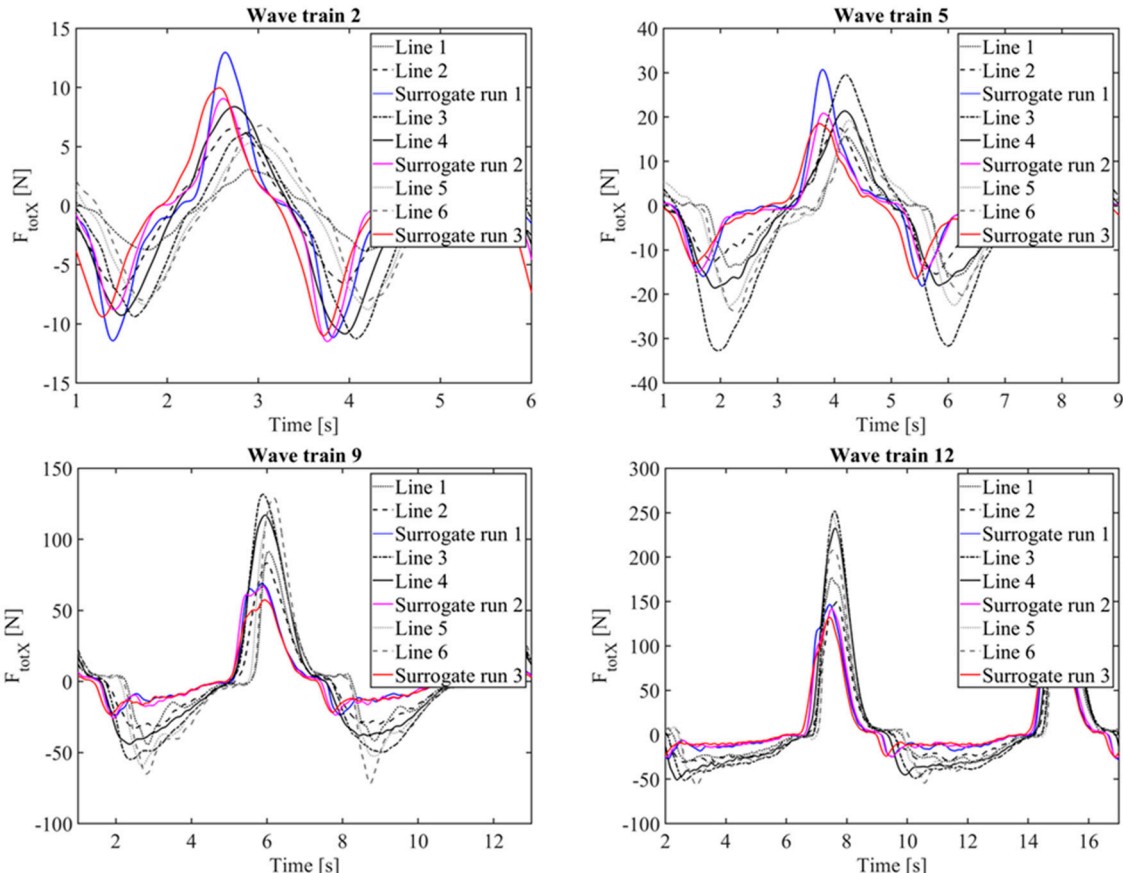

**Figure 10.** Time history of total force in x-direction for example wave cycles of trains 2, 5, 9 and 12.

Force peaks of each individual wave (wave-per-wave maxima) are determined and used to evaluate the force response of the surrogate, as compared to the live dropper lines. The MNE (Equation (2)) between the mean force peaks of all wave trains of the live lines and of the surrogate from the same experimental runs is highest for line 6, with 53.15%, and lowest for line 2, with 11.97%. The RMSE (Equation (3)) is also highest for line 6, with 87.48%, and lowest for line 2, with 28.86%. The highest deviation between individual force peaks is 182.28% for line 6 at wave 9 of wave train 8; the lowest deviation is 0.04% for line 5 at wave 9 of wave train 4. This evaluation of force extrema shows a considerable scatter of the data, since the interaction of wave-induced flows and the ultra-rough surfaces of the shellfish dropper lines is a combined process of turbulence and most likely nonlinear material behavior. Wave-to-wave processes, i.e., out of phase response of the dropper lines, may also contribute to the scatter in MNE and RMSE, and, thus, an averaging approach seems appropriate to summarize and report the relevant forces found in this work. The averaged force peaks of all live lines and all experimental runs of the surrogate, as well as descriptive statistics parameter, are listed in Table 4. Here, starting with wave train 6,

the surrogate underestimates the averaged force peaks of the live lines. The MNE sinks to 22.58% and the RMSE to 48.69% after the averaging approach. Maximum values, as well as standard deviation, are significantly higher for the live lines for higher wave trains.

**Table 4.** Force peaks of total force in x-direction averaged over all live lines and over all runs of the surrogate.

| Wave Train | Live Lines | | | | Surrogate | | | |
|---|---|---|---|---|---|---|---|---|
| | No. of Waves | $F_{max}$ (N) | Mean $F_{max}$ (N) | Std $F_{max}$ (m) | No. of Waves | $F_{max}$ (N) | Mean $F_{max}$ (m) | Std $F_{max}$ (N) |
| 1 | 168 | 7.39 | 4.70 | 1.33 | 84 | 17.28 | 9.23 | 3.39 |
| 2 | 168 | 9.44 | 5.86 | 1.79 | 84 | 12.97 | 10.00 | 0.98 |
| 3 | 138 | 29.79 | 17.83 | 4.58 | 69 | 34.72 | 23.89 | 3.87 |
| 4 | 108 | 34.09 | 18.56 | 5.12 | 54 | 28.07 | 22.04 | 3.42 |
| 5 | 90 | 36.75 | 18.61 | 4.82 | 45 | 30.70 | 20.29 | 3.84 |
| 6 | 78 | 83.11 | 52.42 | 11.76 | 39 | 56.63 | 42.33 | 7.17 |
| 7 | 66 | 80.26 | 56.43 | 12.66 | 33 | 53.32 | 40.40 | 7.71 |
| 8 | 60 | 83.82 | 60.66 | 11.51 | 30 | 47.56 | 36.57 | 7.06 |
| 9 | 48 | 136.68 | 104.73 | 17.52 | 24 | 70.12 | 58.19 | 6.58 |
| 10 | 48 | 135.13 | 103.83 | 19.36 | 24 | 75.61 | 63.12 | 7.52 |
| 11 | 43 | 162.79 | 124.04 | 24.47 | 31 | 94.25 | 77.25 | 7.63 |
| 12 | 36 | 245.47 | 207.51 | 34.24 | 18 | 147.68 | 133.15 | 7.91 |
| 13 | 36 | 244.23 | 190.11 | 32.85 | 18 | 146.83 | 132.65 | 8.62 |

*3.5. Force Coefficients*

All computed drag coefficients of the nine-month-old live lines over *KC* number are depicted in Figure 11, as derived from the wave-per-wave analysis and all experimental runs that were conducted. As expected, values for the live dropper lines scatter over a range of $C_D$ values, and this is mostly attributed to biological irregularities. The values for the surrogate model are generally lower and scatter less. Over the whole *KC* range from 10 to 380, the mean drag coefficient for the live dropper lines is found to be $\underline{C_D}$ = 3.9 ± 2.2, and for the surrogate, $C_D$ = 3.4 ± 2.1.

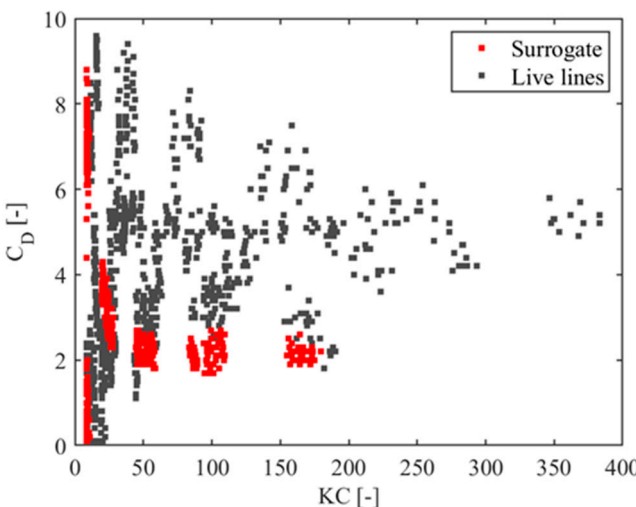

**Figure 11.** Drag coefficients related to the Keulegan–Carpenter (*KC*) number for the live lines and the surrogate.

The mean RMSE (Equation (3)) of the measured and MOJS approximated force peaks, using the experimentally determined drag coefficients, are listed exemplarily for line 3 and the surrogate in Table 5. RMSE decreases from wave train 1 to wave train 13 for all lines; RMSE of force peaks of wave train 1 is 3.1 to 9.5 times higher than the RMSE of force

peaks due to wave train 13. Examples of the force time history of the smallest and highest wave trains in these experiments are depicted for the dropper line 3 in Figure 12. RMSE for the force peaks of wave train 1 is 85.7% and 13% for wave train 13. There is no significant difference between mean RMSE of the live mussel dropper lines and the surrogate with 33.3% and 24.6%, respectively.

**Table 5.** Mean root mean square error (RMSE) of measured and MOJS approximated force peaks of total force in x-direction for Line 3 and surrogate for all wave trains.

| Wave Train | 1 | 2 | 3 | 4 | 5 | 6 | 7 | 8 | 9 | 10 | 11 | 12 | 13 | Mean |
|---|---|---|---|---|---|---|---|---|---|---|---|---|---|---|
| Line 3 RMSE (%) | 85.7 | 70.7 | 35.7 | 20.6 | 11.6 | 13.7 | 16.2 | 14.4 | 12.4 | 8.1 | 18.3 | 18.5 | 13.0 | 26.1 |
| Surrogate RMSE (%) | 66.8 | 25.3 | 32.0 | 32.4 | 32.5 | 30.5 | 28.2 | 21.5 | 8.3 | 9.6 | 12.9 | 11.6 | 8.2 | 24.6 |

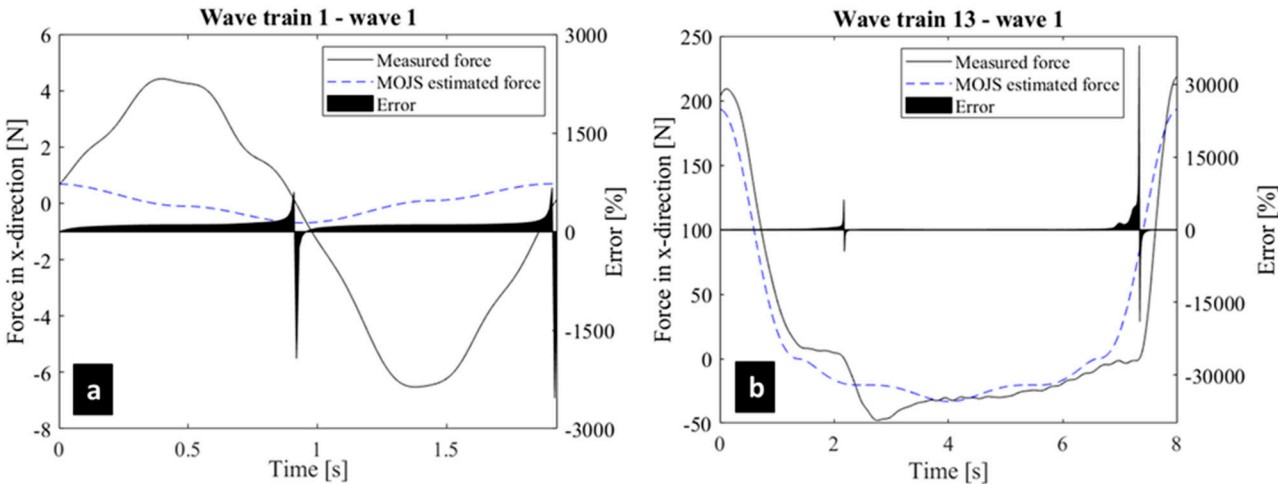

**Figure 12.** Force time histories of example wave cycles, comparing measured and MOJS approximated force in x-direction for line 3, (**a**) wave train 1 and wave 1, and (**b**) wave train 13 and wave 1.

The experimentally determined drag coefficients are next compared to the drag coefficients of rough cylinders in waves [78], overgrown cylinders in oscillating flow [79], both with relative roughness $k/D = 0.02$, and the surrogate in smaller waves [53] (Figure 13). The drag coefficients of rough and overgrown piles are comparable to the coefficients of the surrogate for *KC* numbers between 25 and 60, where mean drag coefficient of the surrogate is $2.4 \pm 0.3$. The drag coefficients of live lines and the surrogate of [53] were extended in this work above *KC* numbers greater than 10. Mean drag coefficients of the surrogate for *KC* numbers below 10 are very well comparable to the coefficients in the *KC* range from 25 to 380 of this work, with 2.4 and 2.6, respectively, whereas the mean drag coefficient of the live lines is significantly higher for higher *KC* ranges. One similarity of all datasets seems to be decreasing scatter with increasing *KC* number, whereas scatter is highest for the live mussel lines and smallest for the overgrown piles. As the live dropper lines and the surrogate model used in this work have considerably higher *h/D* values, it seems clear that these ultra-rough surfaces yield larger forces, indicated by the increase in $C_D$-values. While for closed cylinder surfaces with some surface roughness, the flow around these cylinders will be governed by the overall circular circumference, for the naturally rough surfaces much more interaction, turbulence production and pressure gradients within the vicinity of the structure will occur; this is overall indicated by the increase force that was observed. Future research will have to provide a better picture of the specific hydrodynamic flow field close to these ultra-rough surfaces that was previously unaddressed in the literature.

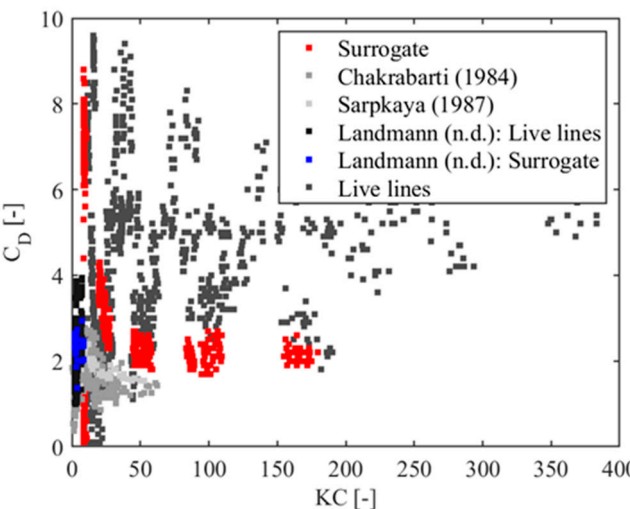

**Figure 13.** Comparison of drag coefficients to rough piles [78], overgrown piles [79] and the surrogate in smaller waves [53].

## 4. Discussion

This work aimed at the quantification of drag coefficients of live bivalve-encrusted dropper lines in waves in large, near full-scale experiments, as well as at the comparison of their force response to a surrogate specimen. As a result of the nature of the live dropper lines, a relative motion, a deflection in the horizontal direction, of the dropper line was observed. This was the inevitable result of the delicate clamping mechanism used to hold the dropper lines in place, and the difficulty to apply vertical pretension, as a too-large pretension would have resulted in excessive mussel drop-off. Due to that relative horizontal motion of the dropper lines, further enhancement of the evaluation method is required for a more precise estimation of drag coefficients, but more importantly of inertia coefficients, $C_M$ [51,52], which will be presented in an upcoming publication. As the relative motion, and thus relative velocity of the fluid field driving the force response is decreased through the elastic behavior of the dropper lines, differences in force coefficients, as well as measured force peaks, are expected; however, the determination of the maximum deflection amplitude was not determined as optical access in the Large Wave Flume and is greatly reduced through suspended sediments.

Drag coefficients of all nine-month-old live lines show high scatter in low *KC* range, additionally underlined by high RMSE (Equation (3)) of force peaks of measured and approximated force. This is connected to the relative motion of the lines and the resulting phase shift between water-level elevation and force in x-direction. The phase-shift decreases with increasing *KC* number since fast waves (high *KC* number) accelerate the lines faster to the maximum point of deflection than slow waves (low *KC* number), leading to decreasing RMSE. Since the drag force is also more dominant at higher *KC* numbers, experimentally determined drag coefficients for low *KC* numbers are suggested to be used carefully only, bearing in mind how these were obtained. Drag coefficients of higher *KC* numbers show the expected scatter due to biological variabilities, stated in high ranges of density, weight per meter and diameter, from live line to live line, but also along an individual line itself. The most decisive factor for the wide scatter could be varying weight in-between the live lines sample, since weight is not included in the MOJS equation, which is assuming rigid objects in waves. As describes above, in these large scale experiments, the mussel lines are flexible and thereof accelerated to a maximum point of deflection in each wave, where measured force results from mass times acceleration. Since acceleration is dependent on drag and is not measured in these experiments, in this publication, no precise description can be given for the correlation of weight of live lines and their estimate drag coefficients. However, one indication might be the very similar drag coefficients of lines 5 and 6 with similar weight, but also similar equivalent diameter. Future tests could ensure higher stiffness by

shortening of the wire ropes between lines and test rack or inserting metal rods into the line before growing mussels, to exclude mass as a factor. The effect of higher stiffness can be seen at the results of live lines and the surrogate of [53], where less scatter occurs for the drag coefficients of the live lines. However, the tests presented in this work highlight many of the effects dropper lines would also experience in natural conditions, and, thus, they provide a first insight into the dynamics relevant for future design approaches.

Lines 5 and 6 also show the lowest values of drag coefficients, which might be a result of highest equivalent diameter, as [53] describe small changes in that parameter as trigger for large changes in the values of force coefficients. Instead, [53] suggest to use a mean diameter in the force coefficient evaluation, which would lead to a decrease of all drag coefficients of live lines in this work. Moreover, important to consider is the effect of mussel drop-off, meaning decreased volume and diameter, which could not be considered in the evaluation. Hence, at high *KC* numbers, the drag coefficients are potentially overestimated, posing economic questions in a design process due to overestimation of the forces. Volume and diameter determination after each wave train in future tests allows for adaptation of the equivalent diameter and avoiding overestimation of the drag coefficient.

The surrogate shows less scatter and lower values for measured force peaks and drag coefficients than the individual live lines, both underlining the previous assumptions of mass and distribution effects of live dropper lines. Although the surrogate was created by imitating the surface of live mussel lines, it has higher stiffness and a regular shape with lower roughness, whereas live dropper lines are ultra-rough due to soft, flexible growth, as algae, anemones, seaweed and sponges, therefore inducing higher drag forces [80]. That high roughness affects the surrounding flow regime as vortex shedding, the interaction of vortices, the separation angle, the turbulence level and the vortex strength [81,82]. This argumentation is supported by similar behavior of drag coefficients for live lines and the surrogate in smaller waves [53], as well as the comparison of the estimated drag coefficients of live mussel dropper lines to rough [78] and overgrown [79] cylinders, also covered with lower roughness and without soft, flexible growth. In the given *KC* range, drag coefficients of the mussel lines are significantly higher than of rough or overgrown cylinder, whereas the best comparability is given by the surrogate. For the sake of completeness, it should be also mentioned that [73] found no influence on drag by mussel feeding; however, feeding is excluded here, since the tests were conducted with saltwater mussels in a freshwater flume.

Last, results for attachment strength, as well as individual mussel length, width and height, are in good agreement with the literature. However, for future work, a horizontal setup is suggested to minimize weakening of the mussel threads during assembly, as in [66]. Regarding future experiments with live mussel dropper lines, mussel drop-off could not only be quantified to determine actual equivalent or mean diameter for the estimation of force coefficients, but also set into context to adhesive forces.

## 5. Conclusions

Measurements in the Large Wave Flume in Hannover of forces on live blue mussel dropper lines and their surrogate, as well as of wave hydrodynamics, were combined to determine corresponding drag coefficients and to evaluate the comparability of their surrogate. In addition, maximum adhesive forces of individual mussels, including measurements of length, width and height, were determined. A mean drag coefficient is estimated to be $C_D = 3.9$ for nine-months-old live blue mussel dropper lines and $C_D = 3.4$ for their surrogate, valid for KC numbers 10 to 380. Lower measured force and lower drag coefficients of the surrogate are a result of lower roughness. The range for drag coefficients for six live lines scatters widely due to biological irregularities. Overestimation of the drag coefficients for high *KC* numbers has to be considered due to mussel drop-off during the experiments. Measured force time histories are well comparable between live lines and the surrogate. Overall, this work shows a surrogate well able to mimic the dynamic response of live blue mussel dropper lines to equivalent waves. Hence, it can be recommended for application in future experiments and in the absence of live specimens for testing line mussel farming.

The maximum adhesive force of a single mussel was determined to be 10.55 N with a mean of 5.07 N in a sample of 82 pull-off tests. The mean mussel was determined to be 3.04 cm long, 1.69 cm high and 1.25 cm wide in a sample of 102 mussels. The results of this work offer knowledge to improve design tools for offshore aquaculture structures and to facilitate the ultimate goal to sustainably nourish a growing world population.

**Author Contributions:** Conceptualization, N.G. and A.H.; methodology, R.G., C.S. and J.L.; software, C.S. and R.G.; validation, C.S. and R.G.; formal analysis, C.S. and R.G.; investigation, J.L., R.G. and L.F.; resources, N.G. and A.H.; data curation, C.S. and R.G.; writing—original draft preparation, R.G.; writing—review and editing, N.G. and J.L.; visualization, C.S. and R.G.; supervision, N.G. and A.H.; project administration, N.G. and A.H.; funding acquisition, N.G. and A.H. All authors have read and agreed to the published version of the manuscript.

**Funding:** This research received no external funding.

**Data Availability Statement:** The data presented in this study are available on request from the corresponding author.

**Acknowledgments:** This research was supported with funding from the New Zealand Ministry of Business, Innovation and Employment through the Cawthron Institute project CAWX1607. Furthermore, the authors gratefully thank Tim Staufenberger from Kieler Meeresfarm for providing the mussel specimen, as well as Dirk and Daniela Haase from Meerwasseraquaristik Haase for providing cooling and aeration equipment for mussel storage. We acknowledge support by the German Research Foundation and the Open Access Publication Funds of Technische Universität Braunschweig.

**Conflicts of Interest:** The authors declare no conflict of interest.

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
