# Peer review of "Large-Scale Laboratory Experiments on Mussel Dropper Lines in Ocean Surface Waves"

_jmse, doi:10.3390/jmse9010029_

Round 1

Reviewer 1 Report

I understand that estimating drag and inertia coefficients of materials is very important to plan aquaculture facilities. But I do not understand the experimental setup in this study. That is, why authors set only bottom half of dropper lines to be submerged in water. In the longline mussel aquaculture, whole droplines are suspended in water as shown in;

https://www.researchgate.net/publication/238731789_Effects_of_Shellfish_Aquaculture_on_Fish_Habitat_Effets_de_la_Conchyliculture_sur_l%27Habitat_du_Poisson

so I wondered that forces on dropper lines are quite different from actual condition. Explanations on the experimental setup are necessary.

In the mussel aquaculture, bottom ends of dropper lines are not tied on the bottom and they will move as to reduce the acting forces so it is important that authors over-estimate the maximum forces. In addition, if authors justify the over-estimation of values as pre-cautionary idea, mussels used in this study are too small to predict forces and coefficients acting on the lines with adult mussels because they grow to 6-10 cm in length.

All the best,

Author Response

The authors thank the reviewer for the valuable comments; these were answered in the attached file.

Reviewer 2 Report

Innovative research addressing a pertinent topic in the growing field of aquaculture. The work provides a good foundation for additional research related to deep water culture of an economically and ecologically important estuarine/marine invertebrate. The manuscript is clearly and concisely written. There are a few duplications in the bibliography needing repair and the following edits:

1) line 558 - change 'evaluated' to 'evaluate'

2) line 628 - aquaculture is misspelled

3) line 634 - feed is misspelled

4) line 650 - insert space between submerged and mussel

5) line 663 - insert space  between 'for' and 'marine'

6) line 608-609 and line 610-611 - duplicate references (Buck et al 2017 a&b)

7) line 662-664 and line 665-667 - duplicate references (Goseberg et al 2017 a&b)

8) line 724-725 and line 726-727 -duplicate references (Olsen et al 2008 a& b)

9) For the above duplicate references (6-8), need to also change citations throughout document.

10) line 718 - repair 'LubchenkoI, J.'

11) line 721 - insert space between of and aquaculture

12) line 728 - fisheries is misspelled

13) line 723 - remove comma at end of title

14) line 761 - population misspelled

15) line 769 - China misspelled

16) line 775-776 - delete last sentence (instructions)

17)  Italics are needed for genus/species throughout bibliography. Species names should be lower case.

Author Response

Thanks for the valuable review comments, a file outlining the changes made is attached.
